# Inhibitory Effect of Avenanthramides (Avn) on Tyrosinase Activity and Melanogenesis in α-MSH-Activated SK-MEL-2 Cells: In Vitro and In Silico Analysis

**DOI:** 10.3390/ijms22157814

**Published:** 2021-07-22

**Authors:** Jun-Young Park, Hyun-Ju Choi, Tamina Park, Moon-Jo Lee, Hak-Seong Lim, Woong-Suk Yang, Cher-Won Hwang, Daeui Park, Cheorl-Ho Kim

**Affiliations:** 1Molecular and Cellular Glycobiology Unit, Department of Biological Sciences, SungKyunKwan University, 300 Chunchun-Dong, Jangan-Gu, Suwon City 440-746, Korea; wnsdud2057@naver.com (J.-Y.P.); skku2017@skku.edu (H.-J.C.); hakseonglim@naver.com (H.-S.L.); 2Department of Predictive Toxicology, Korea Institute of Toxicology, Daejeon 34114, Korea; tamina.park@kitox.re.kr; 3Department of Human and Environmental Toxicology, University of Science and Technology, Daejeon 34113, Korea; 4Department of Herb Science, Dong-Eui Institute of Technology, Busan 47230, Korea; lmj317@dit.ac.kr; 5Nodaji Co., Ltd., Pohang 37927, Korea; yangws91@naver.com; 6Department of AGEE, Handong University, Pohang 37554, Korea; chowon@handong.edu

**Keywords:** Avn-A-B-C, anti-melanogenesis, docking simulation

## Abstract

Melanin causes melasma, freckles, age spots, and chloasma. Anti-melanogenic agents can prevent disease-related hyperpigmentation. In the present study, the dose-dependent tyrosinase inhibitory activity of Avenanthramide (Avn)-A-B-C was demonstrated, and 100 µM Avn-A-B-C produced the strongest competitive inhibition against inter-cellular tyrosinase and melanin synthesis. Avn-A-B-C inhibits the expression of melanogenesis-related proteins, such as TRP1 and 2. Molecular docking simulation revealed that AvnC (−7.6 kcal/mol) had a higher binding affinity for tyrosinase than AvnA (−7.3 kcal/mol) and AvnB (−6.8 kcal/mol). AvnC was predicted to interact with tyrosinase through two hydrogen bonds at Ser360 (distance: 2.7 Å) and Asn364 (distance: 2.6 Å). In addition, AvnB and AvnC were predicted to be skin non-sensitizers in mammals by the Derek Nexus Quantitative Structure–Activity Relationship system.

## 1. Introduction

Oats are known to be rich in water-soluble dietary fiber, protein, vitamins, and minerals [1,2]. In particular, Avenanthramide (Avn), phenolic acid, and flavonoids are the most important phenolic compounds in oats. Phenolic compounds reportedly exhibit a variety of biological activities, such as anti-allergic, antioxidant, anti-inflammatory, and anti-carcinogenic activity [3,4]. The phenolic compounds AvnA, AvnB, and AvnC have been reported to have many anti-oxidant and anti-inflammatory properties, but their role in skin whitening is unknown.

Melanin protects the skin from ultraviolet irradiation-induced oxidative stress, wrinkling, and DNA damage [5]. However, overproduction of melanin causes conditions including hyperpigmentation, freckling, melasma, solar lentigo, and age spots [6]. Sun protection should be the first consideration of any trials regarding hyperpigmentation treatment. Alternative drugs which can replace corticosteroids for melasma or post-inflammatory hyperpigmentation include azelaic acid, retinoids, hydroquinone, and cosmeceuticals [7]. Melanin is present in three different forms, eumelanin (the most abundant), pheomelanin, and neuromelanin [8]. Eumelanin is present in large quantities in black and brown skin in humans [9]. Synthesis of melanin is catalyzed by tyrosinase, forming intermediates such as L-3,4-dihydroxyphenylalanine (L-DOPA) and DOPAquinone from tyrosine as a substrate. Peomelanin and eumelanin are the two reactants formed from DOPAquinone. The known melanogenic enzymes are tyrosinase-related protein-1 and -2 (TRP 1-2) and tyrosinase. Multiple melanogenic enzymes are stored in the endosomal melanosomes of melanocytes for epidemic melanogenesis [10]. Among such enzymes, tyrosinase converts L-tyrosine to the end product dopaquinone via the 3,4-dihydroxyphenylalanine intermediate [11]. Inhibition of melanin biosynthesis prevents melanogenesis; this can be accomplished via specific inhibitors in skincare and medications [12]. Melanogenic inhibitors have been developed that aim to inhibit melanogenic enzymes and the melanin synthetic pathway. Tyrosinase is involved in melanogenesis and tyrosinase inhibitors have been identified in both natural products and chemical synthetics. Tyrosinase inactivators and reversible inhibitors, which are true inhibitors, bind to the enzyme. Screening of tyrosinase inhibitors uses the monophenolic substrate, tyrosine, or the diphenolic substrate, L-DOPA, to examine dopachrome genesis. Tyrosinase is a typical binuclear copper-binding protein. Thus, copper chelating agents can be used as tyrosinase inhibitors and various anti-melanogenic compounds have targeted copper-chelating activities [13]. In addition, the realm of anti-melanogenic cosmetics is interested in dermal safe skin-whitening compounds. As natural tyrosinase inhibitors, phenolic compounds, which have aromatic rings and hydroxyl groups, are isolated from a variety of plants. The compounds are conjugated to saccharides and organic acids. The active ingredients in cosmetics include several whitening agents, such as arbutin [14], hydroquinone [15], deoxyarbutin [16], resorcinol [17], vanillin [18], niacinamide [19], and kojic acid [20]. Isotachioside is an analog of arbutin. Hydroquinone and its derivatives α- and β-arbutin are potent tyrosinase suppressants [21]. Deoxyarbutin and its derivatives are used to lighten hyperpigmented skin lesions, and their effectiveness has been demonstrated. Kojic acid is another known tyrosinase suppressant [22].

Consumers demand accurate estimations of the skin sensitization potential and potency of a cosmetic compound. In addition, animal experiments should be replaced by in silico models and non-animal models of skin sensitization [23,24]. Thus, computational modeling approaches have been used for skin sensitization prediction since 2013 [23]. More recently, quantitative structure–activity relationship (QSAR) approaches with hybrid or integrated models that rely on digital methods have been developed with experimental data using Derek Nexus software. The present study used Derek Nexus to estimate the skin sensitization potential of Avn-A-B-C. AvnB and AvnC were predicted to be non-sensitizers by the Derek Nexus Quantitative Structure–Activity Relationship system.

Avn-A-B-C inhibit melanin synthesis to produce an anti-melanogenic effect in α-MSH-activated SK-MEL-2 cells. Docking simulation and enzyme kinetics data revealed significant information about the active site-binding of Avn-A-B-C. The present results suggest that treatment with Avn-A-B-C inhibits IBMX-induced melanogenesis and tyrosinase activity in SK-MEL2 cells.

## 2. Results

### 2.1. Effects of Avenanthramides (A, B, C) on Cell Viability in α-MSH-Activated SK-MEL-2 Cells

To confirm the cytotoxicity of AvnA, AvnB, and AvnC, an MTT assay was conducted. The none (α-MSH only) group, treated with α-MSH only, showed no significant difference in cell viability compared to the control group. Treatment of cells with AvnA, AvnB, and AvnC in concentrations of 0, 50, and 100 μM produced no cytotoxicity, as the cell viability of the compound-treated groups was not significantly different from that of the None group (Figure 1). AvnA, AvnB, and AvnC at concentrations up to 100 μM did not affect the viability of SK-MEL-2 cells. Thus, these doses were used in further experiments.

### 2.2. Inhibitory Effect of Avenanthramides (A, B, C) on Tyrosinase Activity In Vitro and In α-MSH-Activated SK-MEL-2 Cells

As shown in Figure 2A, in vitro treatment with AvnA, AvnB, and AvnC inhibited tyrosinase activity in a concentration-dependent manner. Cells were treated with compounds at 0, 10, 25, 50, and 100 μM. AvnC had the strongest inhibitory activity, followed by AvnA and AvnB. In addition, α-MSH-induced SK-MEL-2 cells treated with AvnA, AvnB, and AvnC likewise demonstrate concentration-dependent (0, 50, 100 μM) inhibition of tyrosinase activity (Figure 2B). Further experiments in SK-MEL-2 cells showed that AvnC inhibited enzymatic activity to a greater extent than AvnA and AvnB. These data suggest that AvnA, AvnB, and AvnC suppress tyrosinase activity in α-MSH (100 nM)-induced SK-MEL-2 cells. The observed Avn-A-B-C inhibitory activity was dose-dependent, such that 50 µM Avn-A-B-C resulted in the greatest degree of competitive suppression of intercellular tyrosinase and melanin synthesis.

### 2.3. Inhibitory Effect of Avenanthramides (A, B, C) on Melanin Production in α-MSH-Activated SK-MEL-2 Cells

Tyrosinase regulation affects melanin synthesis. Tyrosinase is an enzyme that is involved in melanin biosynthesis; it converts L-DOPA to DOPAquinone and L-tyrosine to L-DOPA to synthesize melanin in biological systems [25,26]. The data shown in Figure 3A, B confirm that melanin production was inhibited by treatment with AvnA, AvnB, and AvnC at every concentration tested (0, 50, 100 μM). AvnC treatment resulted in the strongest inhibition, followed by AvnA and AvnB. These data suggest that the compounds AvnA, AvnB, and AvnC inhibit melanin production in α-MSH-activated SK-MEL-2 cells.

### 2.4. Inhibitory Effect of AvnA, AvnB, and AvnC on Melanogenesis-Related Protein Expression and Dendrite Extension in α-MSH-Activated SK-MEL-2 Cells

We next examined the effect of Avn-A-B-C on expression of melanogenesis-related proteins such as TRP-1, TRP-2, MITF, p-ERK, and p-CREB by Western blotting. Avn-A-B-C inhibited the expression of the melanogenesis-related proteins TRP-1 and -2 and MITF (Figure 4A). Also, kojic acid (100 μM) was used as a positive control (Appendix A). Avn-A-B-C attenuated ERK and CREB phosphorylation (Figure 4A).

In cells, dendritic extension is crucial for forming melanosomes and the formed melanosomes are moved through melanocyte dendrites, transferring melanin pigment. This process is important for epidermal photoprotection and maintenance of normal skin color [27,28]. We also examined the effects of Avn-A-B-C on dendrite extension in SK-MEL-2 cells. While α-MSH treatment stimulated dendrite growth, dendrite extension was attenuated by Avn-A-B-C, as shown in Figure 4B. These data show that the compounds AvnA, AvnB, and AvnC inhibit melanogenesis-related protein expression and dendrite extension in SK-MEL-2 cells.

### 2.5. Molecular Docking Simulation of AvnA, AvnB, and AvnC with Human Tyrosinase

Docking simulations indicated that Avn-A-B-C were located at kojic acid binding sites complexed with tyrosinase (Figure 5 and Appendix A). The three phenylpropanoids of each compound were located inside the pocket. The binding affinity of AvnC for tyrosinase (−7.6 kcal/mol) was greater than that of AvnA (−7.3 kcal/mol) and AvnB (−6.8 kcal/mol) according to AutoDock Vina. The docking simulation revealed seven residues (Asp199, His202, Glu345, Ala357, Gln359, Asn364, and Val377) that participated in hydrogen bonds. AvnA bound to tyrosinase with three hydrogen bonds, at Ala357 (distance: 2.92 Å), Asn364 (3.04 Å), and Val377 (3.26 Å). AvnB bound to tyrosinase with two hydrogen bonds, at Asp199 (3.13 Å) and Ala357 (2.71 Å). AvnC bound to tyrosinase with four hydrogen bonds, at His202 (3.26 Å), Glu345 (3.12 Å), Ala357 (2.87 Å), and Gln359 (2.83 Å). The Ala357 residue interacted in common with Avn-A-B-C whereas AvnC had one more hydrogen bond with Gln359 adjacent to the Ala357. In addition, the Avn-A-B-C had seven or eight hydrophobic interactions with tyrosinase.

The four hydrogen bonds between AvnC and tyrosinase resulted in higher binding affinity of AvnC than that of AvnB or AvnA. Especially, the additional hydrogen bond of the Gln359 residue which was located outside the binding pocket seems to help stabilization of the binding of AvnC on tyrosinase.

### 2.6. Derek Nexus for Prediction of Skin Sensitization

AvnB and AvnC were predicted skin non-sensitizers in mammals. This means that there were no structural alerts, examples of skin sensitization, or unclassified or misclassified features when antofine was entered in the QSAR system. However, AvnA was predicted to have the equivocal skin sensitization potential sub-structure of substituted phenol analogs (Appendix A). The phenol sub-structure in AvnA was matched to the phenolic radicals, structural alert. The structural alert described the potential for skin sensitization by a free radical generation, such as the mechanism of phenolic radicals [29].

## 3. Discussion

The main enzyme involved in melanin biosynthesis is tyrosinase. Most whitening agents regulate the function of tyrosinase or the expression of melanin-associated factors via various signaling systems [30,31,32,33]. In this study, Avn-A-B-C inhibited tyrosinase activity and tyrosinase-related protein expression, thus inhibiting α-MSH-induced melanin production in SK-MEL-2 cells (Figure 6). The results indicated an absence of cytotoxicity (no difference in cell viability between the treatment groups and the ‘None’ group). AvnA, AvnB, and AvnC treatment at concentrations up to 100 μM did not affect the viability of SK-MEL-2 cells (Figure 1). MITF controls the transcription of the three major melanogenic enzymes [34]. Thus, more data is essential to demonstrate the molecular signaling mechanisms through which Avn-A-B-C activate the CREB/MITF pathway. We examined the inhibitory effect of Avenanthramides (A, B, C) on tyrosinase activity and melanin production in vitro and in α-MSH-activated SK-MEL-2 cells. AvnC had the strongest inhibitory activity, followed by AvnA and AvnB (Figure 2 and Figure 3). Furthermore, the effects of Avn-A-B-C treatment on the expression of melanogenesis-related proteins including TRP-1, TRP-2, MITF, p-ERK, and p-CREB were measured. The results indicate that AvnA, AvnB, and AvnC treatment inhibit melanogenesis-related protein expression and dendrite extension in SK-MEL-2 cells (Figure 4).

Since the prohibition on animal testing of cosmetics took effect in 2013 [23], non-animal and in silico models have been applied in the cosmetics field [23,24]. Using QSAR technology, two combined classification models were used for the predicted compound categorization according to skin sensitization potential and regression was done for quantitative potency prediction; all of this was done by Derek Nexus [24]. The models showed that Avn B and Avn C were predicted to be non-sensitizers, i.e., to have no unclassified or unsorted features for skin sensitization in mammals. However, Avn A had a substituted phenol, including a fused ring bond, between the para positions and meta. This alert outlines the skin sensitization potential of the candidate phenol analogs. Skin sensitization results have previously been produced for a few sub-derivatives of phenol, such as 2,5-dimethyl phenol, 3,4-dimethyl phenol [35], and pentachlorophenol [36].

The docking simulation analysis has revealed that AvnC interacted with four residues (His202, Glu345, Ala357, and Gln359) forming hydrogen bonds, while AvnB and AvnA was predicted to form hydrogen bond interactions with three (Ala357, Asn364, and Val377) and two residues (Asp199 and Ala357) of tyrosinase. The four hydrogen bonding interactions between tyrosinase and AvnC could inhibit the tyrosinase activity more effectively. In addition, the Avn-A-B-C had seven or eight hydrophobic interactions with tyrosinase. Furthermore, the docking simulation agrees with the enzyme inhibitory effect and melanin production of Avn-A-B-C (Figure 5).

## 4. Materials and Methods

### 4.1. Reagents

Avn A, B, C were purchased from Sigma-Aldrich (Saint-Louis, MO, USA). Corning trans-well Wright-Giemsa stain solution, 6-diamidino-2-phenylindole dihydrochloride (DAPI), tyrosinase from mushrooms, α-Melanocytes, and L-DOPA (L-3,4-dihydroxyphenylalanine) were obtained from Sigma-Aldrich (Missouri, Saint-Louis, MO, USA). Antibodies against TRP-1 (cat. no. sc-166857; 1:2000), TRP-2 (cat. no. sc-74439; 1:2000), β-actin (cat. no. sc-47778; 1:2000), MITF (cat.no. sc-56725; 1:1000), p-CREB (cat. no. sc-81486; 1:2000), and CREB (cat.no.sc-377154; 1:2000) were obtained from Santa Cruz Biotechnology (California, Santa Cruz, CA, USA). Antibodies against p-ERK (cat. no. 9101S; 1:1000) and ERK (cat. no. 9101S; 1:1000) were obtained from Cell Signaling Technology (Massachusetts, Beverly, MA, USA). Arbutin, Kojic acid, N-phenylthiourea (PTU), 3-isobutyl-1-methylxanthine (IBMX), L-tyrosine, dimethyl sulfoxide (DMSO), and Tween-20 were obtained from Sigma Co. (Missouri, St. Louis, MO, USA). Chemical structures were drawn using Marvin Sketch (https://chemaxon.com/ accessed on 2 August 2020).

### 4.2. Cell Culture

SK-MEL-2 cells were obtained from the American Type Culture Collection (ATCC; Rockville, MD, USA). The cells were incubated in Dulbecco’s modified Eagle medium (DMEM; WelGENE Co., Daegu, Korea) supplemented with 10% fetal bovine serum (FBS), 100 U/mL penicillin, and 100 ng/mL streptomycin and incubated at 37 °C under 5% CO_2_.

### 4.3. Measurement of Tyrosinase Inhibitory Activity

Tyrosinase activity was spectrophotometrically determined as follows: 0.1 mL of 110 U/mL mushroom tyrosinase (EC 1.14.18.1) was added to a mixture of 0.1 mL of 10 mM L-DOPA and 0.05 mL of sample solution in 0.25 mL of 175 mM sodium phosphate buffer (pH 6.8) and reacted at 37 °C for 2 min. The resulting DOPAchrome was measured at a wavelength of 475 nm. The tyrosinase activity inhibition rate was expressed as the rate of decrease in absorbance of the sample solution with and without the addition port. SK-MEL-2 cells were seeded in 24-well plates at a concentration of 2 × 10^5^ cells per well, treated with Avenanthramides, and cultured for 24 h. Then, each well was washed with 10 mM PBS and suspended in 100 μL of 10 mM PBS (1% Triton X-100). After vortexing, the solution was centrifuged and the supernatant was harvested to measure enzyme activity. This enzyme solution (40 µL) was added to a 96-well microplate (BD Falcon, Bedford, MA, USA), and 100 µL L-DOPA (2 mg/mL) was added as the substrate. After the reaction proceeded at 37 °C for 1 h, absorbance was measured at 405 nm using a microplate reader (USA). Inhibitory activity was then calculated using the following equation:Inhibitory activity (%) = [(A − B) − (C − D)]/(A − B) × 100
where A is the absorbance of the control treatment after the reaction, B is the absorbance of the control treatment before the reaction, C is the absorbance of the sample treatment after the reaction, and D is the absorbance of the sample treatment before the reaction. For the tyrosinase kinetic mechanism, various concentrations of L-DOPA (0.625 to 5 mM), mushroom tyrosinase solution (1000 U/mL), and 50 mM potassium phosphate buffer (pH 6.5) were added to a 96-well plate in a total volume of 200 μL, in the presence or absence of Avn-A-B-C. The Michaelis constant (Km) of tyrosinase activity was determined using a Lineweaver–Burk plot with various L-DOPA concentrations.

### 4.4. Measurement of Melanin Production

SK-MEL-2 cells were seeded in a 24-well plate at 2 × 10^5^ cells per well and pretreated with concentrated Avn-A-B-C. After 1 h, α-MSH (100 nM) was added and the cells were incubated for 48 h. Each well was washed with PBS. Then, 400 μL0.2 N NaOH was added and the reagents were dissolved at 60 °C for 1 h. Finally, absorbance was measured at 405 nm.

### 4.5. Western Blotting

We extracted protein from the SK-MEL-2 cells using 1% NP40 protein lysis buffer containing 100 mM Na-Northovanadate, 100 mM NaF, and 100 mM Na-pyrophosphate, and centrifuged the extract at 4 °C at 13,000 rpm. The extracted protein was quantified by the Bio-Rad protein assay (Bio-Rad Laboratories Hercules, CA, USA) and separated the same amount of protein in each sample through SDS-PAGE (SDS-polyacrylamide). Then, we transferred the protein to a NC (nitrocellulose) membrane, reacted it with the primary antibodies and horseradish peroxidase-linked anti-mouse, rabbit, and goat immunoglobulin-G secondary antibodies (Invitrogen and Thermo), and checked the results through ECL (enhanced chemiluminesence).

### 4.6. Docking Simulations of the Compounds with Tyrosinase

The three-dimensional structure of tyrosinase was generated with tyrosinase-related protein 1 (PDB ID: 5M8S) [37] in SWISS-MODEL (https://swissmodel.expasy.org/ accessed on 3 November 2020), which is a homology modeling technique. To prepare AvnA-B-C for the docking simulation, the two-dimensional (2D) structures of the compounds were generated and converted into three-dimensional (3D) structures. Energy minimization was also conducted with the ChemOffice program (http://www.cambridgesoft.com, version: 7.0). The docking simulations (AvnA-B-C to tyrosinase) were performed with Autodock Vina 1.1.2 [38]. The binding pocket of tyrosinase was defined using predefined active sites obtained from the tyrosinase complex with kojic acid (PDB ID: 5M8M) [39]. The results of docking simulations were displayed in Chimera 1.14 [40]. Based on the results of docking simulation, possible hydrogen bonds and hydrophobic interactions using HBPLUS and non-bonded contact parameters as default settings in LigPlot [41].

### 4.7. Derek Nexus for Prediction of Skin Sensitization

The skin sensitization effect of Avn-A-B-C was predicted by Derek Nexus software, which is an expert knowledge-based QSAR system that predicts the potential toxicity of most toxicological endpoints. Nexus version 2.3 including Derek Nexus 6.1 and Sarah Nexus 3.1 (Lhasa Limited, Leeds, Yorkshire, UK) were used to predict the skin sensitization potential of antofine. The toxicity prediction results were assigned to 1 of 17 different categories depending on the weight of the available evidence. If the results classified a test substance as ‘certain’, ‘probable’, ‘plausible’, or ‘equivocal’ it was considered a skin sensitizer. Otherwise, the substance was considered non-toxic.

## 5. Conclusions

Avn-A-B-C produced melanogenic suppression effects in SK-MEL-2 cells by inhibit-ing intracellular tyrosinase activity. Avn-A-B-C treatment in IBMX-stimulated melanin biosynthesis cells led to downregulation of the expression levels of melanin syn-thesis-related mRNAs and proteins. These results indicate that Avn-A-B-C act as melano-genesis-suppression agents with dual functions, blocking the generation of melanogene-sis-associated factors and inhibiting the tyrosinase enzymatic response. According to the QSAR analysis, skin sensitization is very unlikely. Therefore, the sanitary-safety analysis demonstrated that Avn-A-B-C can be used for cosmetic purposes or skin protection. The molecular docking simulation showed that AvnC bound most strongly to tyrosinase, with a binding affinity of −7.6 kcal/mol. This value was greater than that observed for AvnA (−7.3 kcal/mol) and AvnB (−6.8 kcal/mol). These results support the notion that compound AvnC should be considered the most potent and selective tyrosinase inhibitor.

## Figures and Tables

**Figure 1 ijms-22-07814-f001:**
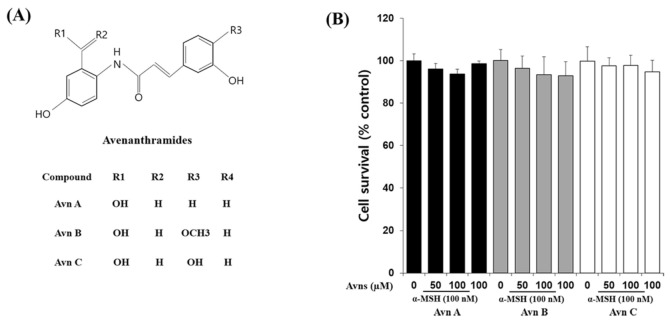
Effects of avenanthramides (A, B, C) on cell viability in α-MSH-activated SK-MEL-2 cells. (**A**) Avenanthramides (**A**–**C**) (chemical structure). (**B**) Cellular survivability was assessed using MTT assay. Treatment of cells with AvnA, AvnB, and AvnC in concentrations of 0, 50, and 100 μM produced no cytotoxicity, as the cell viability of the compound-treated groups was not significantly different from that of the none group.

**Figure 2 ijms-22-07814-f002:**
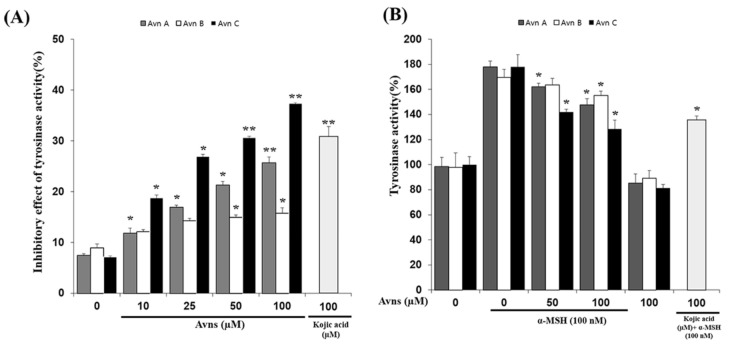
Inhibitory effect of Avenanthramides (A, B, C) on tyrosinase activity in vitro and in α-MSH-activated SK-MEL-2 cells. (**A**) in vitro treatment with AvnA, AvnB, and AvnC inhibited tyrosinase activity in a concentration-dependent manner. Cells were treated with compounds at 0, 10, 25, 50, and 100 μM. Kojic acid (100 μM) was used as a positive control. (**B**) α-MSH-induced SK-MEL-2 cells treated with AvnA, AvnB, and AvnC likewise demonstrate concentration-dependent (0, 50, 100 μM) inhibition of tyrosinase activity. Kojic acid (100 μM) was used as a positive control. The results shown are mean ± SEM and represent three independent tests. * *p* < 0.05 and ** *p* < 0.01 = significant differences from the α-MSH -treated cells.

**Figure 3 ijms-22-07814-f003:**
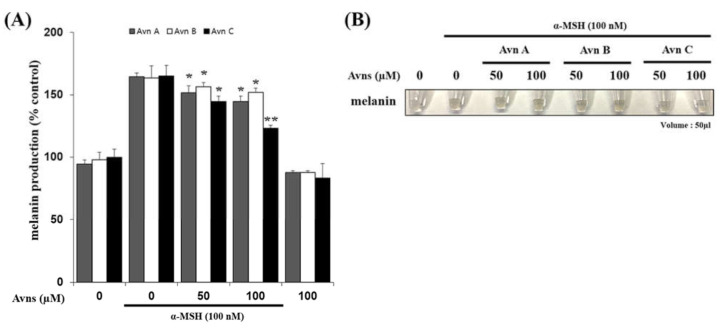
Inhibitory effect of Avenanthramides (A, B, C) on melanin production in α-MSH-activated SK-MEL-2 cells. (**A**) melanin production was inhibited by treatment with AvnA, AvnB, and AvnC at every concentration tested (0, 50, 100 μM). (**B**) melanin production photograph with AvnA, AvnB, and AvnC at concentration tested (0, 50, 100 μM). The results shown are mean ± SEM and represent three independent tests. * *p* < 0.05 and ** *p* < 0.01 = significant differences from the α-MSH treated cells.

**Figure 4 ijms-22-07814-f004:**
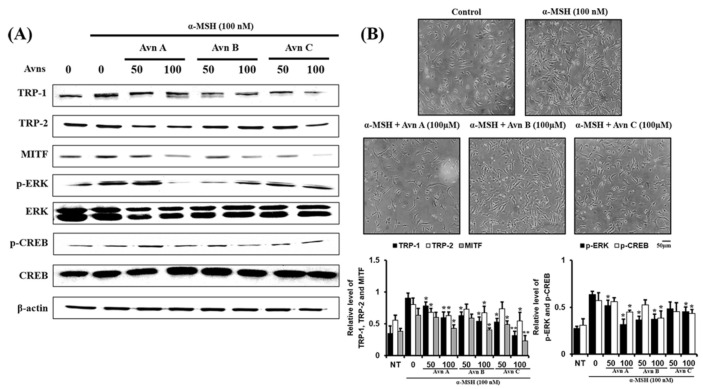
Inhibitory effect of AvnA, AvnB, and AvnC on melanogenesis-related protein expression and dendrite extension in α-MSH-activated SK-MEL-2 cells. (**A**) Avn-A-B-C on expression of melanogenesis-related proteins such as TRP-1, TRP-2, MITF, p-ERK, and p-CREB by Western blotting. Cells were treated with compounds at 0, 10, 25, 50, and 100 μM. (**B**) Cell morphology of micrograph (scale bar is 50 μm). Avn-A-B-C on dendrite extension in SK-MEL-2 cells. Cells were treated with compounds at 100 μM. The results shown are mean ± SEM and represent three independent tests. * *p* < 0.05 and ** *p* < 0.01 = significant differences from the α-MSH treated cells.

**Figure 5 ijms-22-07814-f005:**
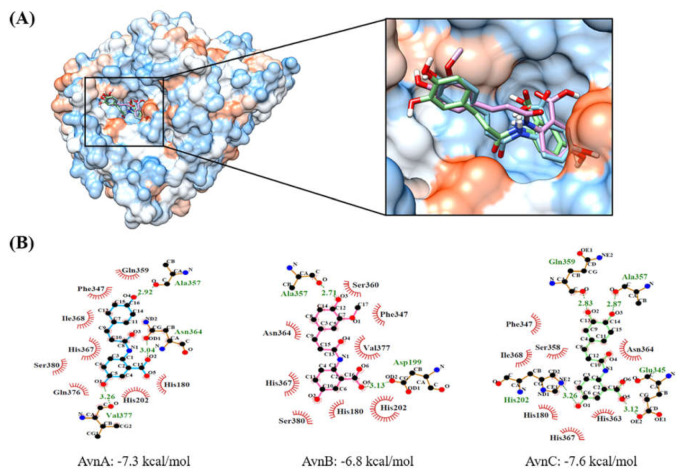
Molecular docking simulation of AvnA, AvnB, and AvnC with human tyrosinase. (**A**) Docking simulations indicated that Avn-A-B-C were located at kojic acid binding sites complexed with tyrosinase. The three phenylpropanoids of each compound were located inside the pocket. The binding affinity of AvnC for tyrosinase (−7.6 kcal/mol) was greater than that of AvnA (−7.3 kcal/mol) and AvnB (−6.8 kcal/mol) according to AutoDock Vina. (**B**) Interaction analysis between tyrosinase and Avn-A-B-C. The Avn-A-B-C had several hydrogen bond interactions at seven residues (Asp199, His202, Glu345, Ala357, Gln359, Asn364, and Val377) on tyrosinase. The four hydrogen bonds between AvnC and tyrosinase resulted in higher binding affinity of AvnC than that AvnB or AvnA. In addition, the Avn-A-B-C had seven or eight hydrophobic interactions with tyrosinase.

**Figure 6 ijms-22-07814-f006:**
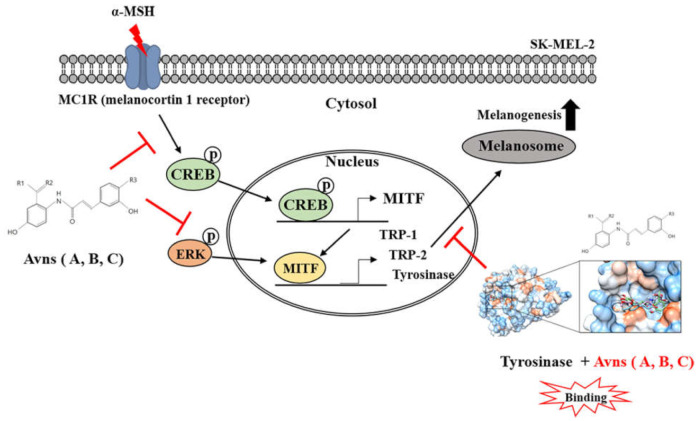
Schematic illustration of inhibitory effect of Avn-A-B-C on tyrosinase activity and melanogenesis in α-MSH-activated SK-MEL-2 cells. Inhibitory effect of Avn-A-B-C on tyrosinase activity and melanogenesis in α-MSH-activated SK-MEL-2 cells: in vitro and in silico analysis. Avn-A-B-C inhibit melanin synthesis to produce an anti-melanogenic effect in α-MSH-activated SK-MEL-2 cells. Docking simulation and enzyme kinetics data revealed significant information about the active site-binding of Avn-A-B-C. Treatment with Avn-A-B-C inhibits IBMX-induced melanogenesis and tyrosinase activity in SK-MEL2 cells.

## Data Availability

Not applicable.

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
