# Peer review of "Inhibitory Effect of Avenanthramides (Avn) on Tyrosinase Activity and Melanogenesis in α-MSH-Activated SK-MEL-2 Cells: In Vitro and In Silico Analysis"

_ijms, 2021, doi:10.3390/ijms22157814_

Round 1

Reviewer 1 Report

This work describes what appears to be a new function of avenanthramides (Avn), and a new solution - and easy to carry out - to combat excess skin pigmentation. If this is so, this work is a very interesting novelty because it also reports what appears to be a new role for these compounds. However, the manuscript should be corrected very deeply - especially and with great importance, the introduction - to present and highlight the main research topic well. The points that should be reviewed are as follows.

Introduction

  1. The authors dedicate much space to the need for effective treatments for excess pigmentation. This can be greatly simplified because it is a well-known topic. Instead, the introduction should be extended to the following four points, which are hardly or not at all commented, with sufficient supporting references.
  2. A discussion of previous treatments that exist to reduce or prevent skin pigmentation.
  3. Since this work is based on tyrosinase being the most critical step in melanin synthesis, this should be discussed.
  4. A brief review of other known tyrosinase inhibitors.
  5. Other previously recognized Avn functions and uses in order to highlight the new found role.

Results

  1. The results with Avn A, B, and C are similar. Delete the sentence at the end of the abstract: "AvnC is a more potent and selective tyrosinase inhibitor than AvnA and -B" and other similar statements.

Title

  1. The title must also be corrected. The Avn abbreviation may not be recognized by a number of readers. It should be rather:

“Inhibitory effect of Avenanthramides (Avn)… (without specifying that they are A-B-C) on tyrosinase activity ... "

Author Response

"Please see the attachment (total Revision answer file)."

Reviewer 2 Report

1. Virtual docking was performed to check the binding of Avn with tyrosinase. However, this might not reflect the actual binding relationship between these two molecules. Chemical analysis of these two molecules should be carried out. 2. Figure 4A: plot a summary quantification graph for the western blot result.   3. Figure 4B: scale of each photo is not shown. 4. The result from Derek Nexus is not shown. A summary of the analysis should be given. 5. Positive control is missing in each experiment. Please suggest a known tyrosinase-inhibiting drug as the positive control. 6. Discuss how dendrite extension affects melanogenesis or hyperpigmentation.  

Author Response

(The authors gave the same response as above.)

Round 2

Reviewer 1 Report

The authors have responded appropriately to all my suggestions. I have no objection to this version. I think the article is ready to be published.

Author Response

The authors have responded appropriately to all my suggestions. I have no objection to this version. I think the article is ready to be published.

The answer: I appreciate the reviewer for careful reading.

Reviewer 2 Report

In this manuscript, the authors aimed to investigate the function of Avn-A-B-C in regulating melanogenesis in α-MSH-activated SK-MEL-2 cells. Methods used included MTT assay to assess the optimal drug concentration, tyrosinase assay and Western blotting to analyse the expression of melanogenesis-related proteins, molecular docking to study the conformational relationship between the compounds and tyrosinase, as well as Derek Nexus Quantitative Structure-Activity Relationship system to predict the drugs sensitization in mammals.

Comments:

Most of the major questions raised in former review have been solved.  

Question 5 in last review: Positive control is missing in each experiment. Please suggest a known tyrosinase-inhibiting drug as the positive control. Positive controls for tyrosinase assay and Western blotting should be added as well.

Author Response

I appreciate the reviewer for careful reading. 
